# Oxidative Stress Response Mechanisms Sustain the Antibacterial and Antioxidant Activity of *Quercus ilex*

**DOI:** 10.3390/plants13081154

**Published:** 2024-04-21

**Authors:** Mariavittoria Verrillo, Piergiorgio Cianciullo, Vincenza Cozzolino, Francesca De Ruberto, Viviana Maresca, Alessia Di Fraia, Lina Fusaro, Fausto Manes, Adriana Basile

**Affiliations:** 1Department of Agricultural Sciences, University of Naples “Federico II”, Piazza Carlo di Borbone 1, 80055 Portici, Italy; mariavittoria.verrillo@unina.it; 2Centro Interdipartimentale di Ricerca per la Risonanza Magnetica Nucleare per l’Ambiente, l’Agroalimentare, ed i Nuovi Materiali (CERMANU), University of Naples “Federico II”, Piazza Carlo di Borbone 1, 80055 Portici, Italy; 3Department of Biology, University Federico II Via Cinthia 26, 80126 Napoli, Italy; piergiorgio.cianciullo@unina.it (P.C.); viviana.maresca@unina.it (V.M.); alessia.difraia@gmail.com (A.D.F.); 4Department of Clinical Medicine and Surgery, University of Naples “Federico II”, Via Pansini, 5, 80131 Naples, Italy; francesca.deruberto@unina.it; 5National Research Council, Institute of BioEconomy, Via dei Taurini 19, 00185 Rome, Italy; lina.fusaro@cnr.it; 6Department of Environmental Biology, Sapienza University of Rome, p.le Aldo Moro 5, 00185 Rome, Italy; fausto.manes@uniroma1.it

**Keywords:** ozone, nitrogen, bioactive compounds, environmental management, antimicrobial efficacy, radical scavenger activity

## Abstract

The development of new natural antibiotics is considered as the heart of several investigations in the nutraceutical field. In this work, leaves of *Quercus ilex* L. treated by tropospheric ozone (O_3_) and nitrogen (N) deposition, exhibited a clear antimicrobial efficacy against five multi-drug resistant (MDR) bacterial strains (two gram-positive and three gram-negative). Under controlled conditions, it was studied how simulated N deposition influences the response to O_3_ and the antibacterial and antioxidant activity, and antioxidant performance. The extraction was performed by ultra-pure acetone using two different steps. A higher antioxidant activity was measured in the presence of interaction between O_3_ and N treatments on *Quercus* leaves. At the same time, all organic extracts tested have shown bacteriostatic activity against all the tested strains with a MIC comprised between 9 and 4 micrograms/mL, and a higher antioxidant efficacy shown by spectrophotometric assay. Stronger antimicrobial activity was found in the samples treated with O_3_, whereas N-treated plants exhibited an intermediate antibacterial performance. This performance is related to the stimulation of the non-enzymatic antioxidant system induced by the oxidative stress, which results in an increase in the production of antimicrobial bioactive compounds.

## 1. Introduction

Antioxidant enzymes are responsible for reactive oxygen species (ROS) neutralization activity [1]. Nevertheless, each subcellular compartment produces its own ROS and has its own set of enzymes, suggesting a large variability within cellular homeostasis [2]. At the same time, antioxidant enzymes such as ascorbate peroxidase (APX), superoxide dismutase (SOD), catalase (CAT), and peroxidase (POD) are involved in ROS deletion [2,3]. ROS production stimulates the detoxifying barrier in the apoplast and enzymatic activity at the symplastic level, which has a high metabolic cost, and the ability to increase antioxidant defenses is identified as a key contributor to ozone tolerance [4]. ROS can be produced by NADPH oxidases, superoxide enzymes, and apoplast peroxidases, and are generally a necessary signal to trigger the stress cascade [5]. ROS have a two-faced Janus role in plant organisms: they serve as signaling agents in many biological processes (e.g., cell proliferation) but on the other hand, their imbalance can cause harmful effects in plant cells such as DNA damage, membrane disruption, and finally, cell death [6]. Plant organisms have evolved different processes to counteract the imbalance of ROS and the subsequent oxidative stress. To maintain the cellular redox homeostasis, plants activate a wide range of cellular responses that collectively take the name of “antioxidant systems”. Recent studies have shown that there is an interaction between tropospheric ozone (O_3_) and nitrogen (N) deposition that results in nitrogen being available for photosynthesis in *Fagus crenata* [7]. Furthermore, the increase in the root system in the presence of nitrogen does not occur when the O_3_ concentration is very high compared to the control in *Quercus robur* L. [8]. In evergreen species as *Quercus ilex* L. (*Q. ilex*), nitrogen is preferentially allocated to the cell walls, leading to an increase in the persistence of leaves [5], as well as toughness and chemical defense [9]. Generally, evergreen species’ response to general abiotic stress included the synthesis of O_3_ by ROS. Due to that feature, evergreen species can be suitable for a study of the mechanism of plant antimicrobial activity, since it is mediated via ROS generation, intracellular protein leakage, and the rupture of the bacterial cell membrane. Although a wide range of papers have focused on the effects that O_3_ has at the leaf, plant, or ecosystem levels [10,11,12], in our knowledge, there are no attempts to focus the attention on the antibacterial potential that the cellular mechanisms overseeing the antioxidant defense deployed by plants can have.

Moreover, under global environmental changes, the exposure of Mediterranean forests to biotic and abiotic stress factors, either individually or in combination, is expected to increase [13]. Stressors affect plant functionality, simultaneously or successively over time, enhancing the oxidative pressure [13]. Among these stressors, tropospheric ozone (O_3_) and anthropogenic reactive nitrogen (N) emissions co-occur and may affect vegetation functions [14]. In particular, O_3_ represents a significant phytotoxic air pollutant [15] that can induce several damages to vegetation by increasing the oxidation load, thereby triggering the production of reactive oxygen species (ROS) that leads to alterations in functional processes at different levels [10,16]. Otherwise, N deposition can have different effects depending on the exposure level: moderate N deposition may favor primary production, increasing pigment content or enzymatic activity, but several studies report that the addition of N increases leaf N content, photosynthetic rate, and thus, the uptake of gases via increased stomatal conductance, amplifying the damages caused by O_3_ [10,11,12].

In this work, we hypothesize that the employment of O_3_ and N could presumably elicit better antibacterial and antioxidant activities in the leaf extracts of the evergreen species *Q. ilex*. We evaluated the potential in vitro antioxidant activity and the antibacterial efficacy of natural extracts against five multi-drug resistant bacterial strains (two gram-positive and three gram-negative) to identify a possible application of these natural products in the nutraceutical field. Additionally, in this work we studied how the nitrogen-induced changes influence the response to O_3_ and, consequently, how nitrogen deposition affects the antibacterial activity of *Q. ilex* extracts. We chose *Q. ilex* as the target species because it has been found to be a tolerant species to oxidative stressors due to different antioxidant response mechanisms [17]. Moreover, due to its wide natural distribution in the Mediterranean Basin, it can be easily employed as source of antioxidant molecules with antibacterial activity.

## 2. Results

### 2.1. Chemical Characterization of Q. ilex Leaves Extracts

#### 2.1.1. Hydroxicinammic Acids

This study investigated the influence of diverse treatments on the concentrations of hydroxycinnamic acids in *Q. ilex* leaves, quantified in milligrams of catechin equivalents per gram (mg/g) of dry weight (mg CE g^−1^ DW) (Figure 1). The mean concentrations of hydroxycinnamic acids, along with their respective standard deviations for each treatment group were as follows: CTRL (28.67 ± 0.58 mg CE g^−1^ DW), O_3_ (56.33 ± 1.76 mg CE g^−1^ DW), N (64.33 ± 2.03 mg CE g^−1^ DW), and O_3_+N (76.33 ± 1.86 mg CE g^−1^ DW). The leaves from plants treated with a combination of O_3_ and nitrogen exhibited the highest levels of hydroxycinnamic acids, followed by the nitrogen treatments. Treatments with O_3_ alone also resulted in a substantial increase in hydroxycinnamic acid content compared to the control group.

#### 2.1.2. Total Flavonoid Content

The total flavonoid content was measured in *Q. ilex* leaves under the various experimental treatments (Figure 1) and is expressed as mg rutin equivalents g^−1^ DW (mg RE g^−1^ DW). The mean total flavonoid contents for each treatment group were: CTRL (34 ± 1.15 mg RE g^−1^ DW), O_3_ (36.33 ± 1.20 mg RE g^−1^ DW), N (45.67 ± 0.67 mg RE/g^−1^ DW), and O_3_+N (59 ± 0.58 mg RE g^−1^ DW). The highest content in total flavonoids was measured in the leaves collected from the plants treated with O_3_ and nitrogen together, followed by the nitrogen alone treatment. Treatment with O_3_ alone also resulted in a noticeable increase in total flavonoid content compared to the control group.

#### 2.1.3. Total Phenolic Content

The total phenolic content was measured in *Q. ilex* leaves’ extracts under the different treatment conditions and is expressed as mg rutin equivalents g^−1^ DW (mg RE g^−1^ DW). The mean total phenolic contents for each treatment group (Figure 1) were as follows: CTRL (65 ± 2.1 mg RE g^−1^ DW), O_3_ (73 ± 2.51 mg RE g^−1^ DW), N (87 ± 2.32 mg RE g^−1^ DW), and O_3_+N (110 ± 3.12 mg RE g^−1^ DW). The combined treatment with O_3_ and N exhibited the highest levels of total phenolic content, followed by the N treatment. The treatments with O_3_ alone also resulted in a notable increase in total phenolic content compared to the control group.

#### 2.1.4. Reduced Glutathione

Glutathione, a crucial antioxidant involved in plant defense mechanisms, plays a pivotal role in mitigating oxidative stress. In this study, we investigated the impact of different treatments on glutathione levels in *Q. ilex* leaves (Figure 2). The mean glutathione levels and standard deviations for each treatment group were as follows: CTRL (258.67 ± 11.2 mmol/g DW), O_3_ (339.33 ± 26.5 mmol g^−1^ DW), N (452.33 ± 27.0 mmol g^−1^ DW), and O_3_+N (481.17 ± 6.5 mmol g^−1^ DW).

The combined treatment of O_3_ and nitrogen resulted in the highest glutathione levels followed by nitrogen treatment alone. Conversely, the control group exhibited the lowest glutathione levels.

#### 2.1.5. Ascorbic Acid

In this study, we investigated the effect of various treatments on ascorbic acid levels in *Q. ilex* leaves (Figure 2). Mean ascorbic acid levels and standard deviations for each treatment group were as follows: CTRL (1.37 ± 0.17 mmol g^−1^ DW), O_3_ (2.65 ± 0.19 mmol g^−1^ DW), N (3.55 ± 0.21 mmol g^−1^ DW), and O_3_+N (3.89 ± 0.24 mmol g^−1^ DW).

The highest levels of ascorbic acid in the leaves were observed when plants were treated with O_3_ and nitrogen together, followed by the N-treated samples (3.55 ± 0.21 mmol g^−1^ DW). Treatment with O_3_ alone also led to a significant increase in ascorbic acid levels (2.65 ± 0.19 mmol g^−1^ DW) compared to the control group (1.37 ± 0.17 mmol g^−1^ DW).

### 2.2. In Vitro Antioxidant Activities of Q. ilex Leaves Extracts

#### 2.2.1. Ferric Reducing Antioxidant Power (FRAP) Assay

The FRAP assay was used to measure the in vitro antioxidant activity of *Q. ilex* leaves extracts under varying treatment conditions, and it was expressed as milligrams of trolox equivalent per gram of DW (mg TE g^−1^ DW) (Figure 3). The mean FRAP values, accompanied by standard deviations for each treatment group, were as follows: CTRL (45 ± 2.03 mg TE g^−1^ DW), O_3_ (54 ± 2.08 mg TE g^−1^ DW), N (67 ± 1.01 mg TE g^−1^ DW), and O_3_+N (72 ± 1.02 mg TE g^−1^ DW). The O_3_+N treatment exhibited the highest FRAP activity followed by the nitrogen treatment. Treatment with ozone alone also resulted in a considerable increase in total antioxidant capacity compared to the control group.

#### 2.2.2. 2,2-Diphenyl-1-picrylhydrazyl (DPPH) Assay

The DPPH assay was measured in *Q. ilex* leaves extracts under the diverse treatment conditions and was expressed as milligrams of trolox equivalent per gram of DW (mg TE/g DW) (Figure 3).

The mean DPPH assay values, accompanied by standard deviations for each treatment group, were as follows: CTRL (35.33 ± 1.53 mg TE g^−1^ DW), O_3_ (44.67 ± 1.53 mg TE g^−1^ DW), N (48.33 ± 0.58 mg TE g^−1^ DW), and O_3_+N (50.33 ± 4.16 mg TE g^−1^ DW). The joint treatment with O_3_ and nitrogen exhibited the highest levels of DPPH activity, followed by the nitrogen alone treatment. Treatment with O_3_ alone also resulted in a notable increase in total antioxidant capacity compared to the control group.

#### 2.2.3. 2,2′-Azino-bis-3-ethylbenzothiazoline-6-sulfonic Acid (ABTS) Assay

The ABTS assay was measured in *Q. ilex* leaves extracts under diverse treatment conditions and was expressed as milligrams of trolox equivalent per gram of DW (mg TE g^−1^ DW) (Figure 3). 

The mean ABTS assay values, accompanied by standard deviations, for each treatment group were as follows: CTRL (632.68 ± 2.30 mg TE g^−1^ DW), O_3_ (856.04 ± 2.7 mg TE g^−1^ DW), N (721.34 ± 2.53 mg TE g^−1^ DW), and O_3_+N (678.39 ± 2.4 mg TE g^−1^ DW). The O_3_ alone treatment exhibited the highest levels of ABTS activity, followed by the nitrogen treatment. The combined treatment with O_3_ and nitrogen also showed notable antioxidant capacity, whereas the control group displayed a lower ABTS activity.

### 2.3. Antibacterial Activity of Q. ilex Leaves Extracts 

The minimum inhibitory concentrations (MICs) of *Q. ilex* leaves extracts were measured after the different treatments and were expressed in micrograms per milliliter (µg/mL) (Figure 4). The assayed bacterial strains comprised Escherichia coli ATCC35218, Staphylococcus aureus ATCC6538P, Pseudomonas aeruginosa ATCC27355, Enterococcus faecalis ATCC29212, and Klebsiella pneumoniae ATCC700503.

For Escherichia coli ATCC35218, the control (CTRL) leaves extracts exhibited a MIC of 9.4 ± 0.2 µg/mL, the highest among the treatments, suggesting limited inherent resistance. Ozone treatment (O_3_) showed promising results with a MIC of 4.033 ± 0.115 µg/mL. Nitrogen (N) and the combined O_3_ and nitrogen (O_3_+N) treatment had MICs of 6.4 ± 0.1 µg/mL and 5.3 ± 0.1 µg/mL, respectively.

In the case of Staphylococcus aureus ATCC6538P, the most effective treatment was O_3_ (2.767 ± 0.058 µg/mL), which notably outperformed the nitrogen treatment (7.200 ± 0.115 µg/mL). The control leaves extracts showed the highest MIC compared to those from the treated plants (8.700 ± 0.112 µg/mL). The combined O_3_ and nitrogen treatment resulted in a MIC (3.000 ± 0.012 µg/mL), positioning it as a strong contender against O_3_ alone.

For Pseudomonas aeruginosa ATCC27355, the O_3_ treatment yielded the lowest MIC (4.567 ± 0.062 µg/mL), displaying enhanced antibacterial activity relative to nitrogen alone (5.000 ± 0.100 µg/mL) and the joint O_3_ and nitrogen treatments (5.100 ± 0.012 µg/mL), with the control displaying the highest MIC (9.600 ± 0.100 µg/mL).

Enterococcus faecalis ATCC29212 demonstrated the greatest sensitivity to leaves extracts from nitrogen-treated plants (3.000 ± 0.104 µg/mL), which was more effective than the O_3_ treatment (4.000 ± 0.102 µg/mL) and the combined O_3_ and nitrogen treatments (6.000 ± 0.100 µg/mL), while the control leaves extracts (7.967 ± 0.061 µg/mL) showed the least efficacy.

Lastly, Klebsiella pneumoniae ATCC700503 exhibited the lowest resistance to leaves extracts from the O_3_-treated plants (3.000 ± 0.100 µg/mL) and the combined O_3_ and nitrogen treatments (5.000 ± 0.113 µg/mL) offered moderate inhibitory effects. The control group yielded a MIC (7.267 ± 0.055 µg/mL) indicative of higher resistance in the absence of treatment.

## 3. Discussion

The antioxidant capacities of *Q. ilex* samples have been reported and are shown in Figure 3. Different spectrophotometer techniques highlight that higher antioxidant activity was measured in the interaction between O_3_ and N treatments whereas plants treated by only N or O_3_ exhibited a moderate antioxidant performance compared to untreated samples (CTRL). A similar trend was observed in Figure 1 and Figure 2 where the total content of flavonoid, hydroxycinnamic acids, phenolic compounds, ascorbic acid, and reduce glutatione has been quantified. *Q. ilex* has been characterized by a slow growth strategy and a conservative pattern of nutrient use [18], but when N is available, as when deposition occurred, the metabolic costs of antioxidant defense are strengthened and higher activation takes place, implying a reduction in the damaging effects of O_3_ on functional traits [10,11]. It is interesting to underline that this response has been observed for realistic doses (20 kg N ha^−1^ yr^−1^ and 80 ppb h for N and O_3_, respectively), making the processes highlighted in this paper comparable to what can happen in nature. Previous works have focused attention on functional traits at the leaf or plant level [12], but only a few have paid attention to biochemical traits; [7,10,19] showed that an elevated content of O_3_ can reduce the root biomass in the presence of nitrogen addition. At the same time, O_3_ could induce a modulation in the synthesis of flavonoids, hydroxycinnamic acids, phenolic compounds, ascorbic acid, and reduce glutatione in plants [20]. Phenolic compounds are the most prevalent secondary metabolites found in plants, constituting approximately 30–45% of the plant’s organic matter. These compounds are primarily synthesized from L-phenylalanine through the nitrogen-free structures of t-cinnamate [21]. Moreover, flavonoids, which are also produced using phenylalanine, may be influenced by nitrogen metabolism. In this context, the initial phase of the phenylpropanoid biosynthesis pathway involves the action of phenylalanine ammonia-lyase (PAL). PAL catalyzes the conversion of L-phenylalanine into trans-cinnamic acid, serving as a precursor for flavonoid synthesis. Ammonium released during PAL activity is assimilated by plants through glutamine synthetase (GS) and glutamine-2-oxoglutarate amidotransferase (GOGAT), subsequently re-entering the phenylpropanoid biosynthetic pathway [22]. Our results may suggest that the treatment of *Q. ilex* leaves by N induces a possible stimulation of biochemical pathways involved in flavonoids production, influencing some enzymatic reaction. Several studies suggest that nitrogen supply can affect flavonoid biosynthesis, i.e., by lowering the activity of the phenylalanine ammonia-lyase (PAL) and thus the flavonoids and phenolic acid contents [23].

Additionally, flavonoids such as phenols or hydroxycinnamic acid are recognized as vegetable secondary metabolites. In this work, the higher content of phenolic compounds in leaves treated with a combination of ozone and nitrogen has been related to the formation of active oxygen radicals in vegetable systems [24]. At the same time, reduced glutathione has been identified as a tripeptide of glutamate, cysteine, and glycine with a role as a precursor of phytochelatins (PC), in redox signaling, ion homeostasis, and sulfur assimilation [25]. Generally, GSH reduces the content of H_2_O_2_ and lipid peroxides via peroxidase-catalyzed reactions. Our data, as reported in Figure 2, confirm that GSH production is increased in ozone treatment but also in combined ozone/nitrogen treatment, which is therefore the best combination for stress tolerance in *Q. ilex* leaves [25]. In this context, free radicals destroy the cell wall and the plasma membrane, and this leads to cell deterioration, loss of photosynthetic efficiency, accelerated leaf senescence, loss of ability to tolerate other stresses, and reduced growth. Although the induction of antioxidant systems has been observed, the role of antioxidants in protecting against O_3_-induced damage is unclear and the results are controversial [26]. The oxidizing nature of ROS could represent a problem with normal cellular action and, if uncontrolled, could induce cell death [27]. Protection from the potential damage to cellular components caused by O_3_ and relative induced ROS is based on the balance between ROS production and elimination at the intracellular level. This balance is effectively carried out by enzymatic and nonenzymatic antioxidants [28]. Plant carbon metabolism in the presence of ozone stress induces the synthesis of compounds such as lipids, organic acids, phenol, and structural carbohydrates, which help plants increase their tolerance/resilience to O_3_ stress [29]. Carbohydrates, proteins, and phenols are important metabolites affected by O_3_ stress. Different studies have confirmed that plants growing under O_3_ stress convert mobile carbohydrates into less mobile secondary metabolites. In this context, under O_3_ stress, plants divert much of their carbon skeleton to pathways that lead to the synthesis of secondary metabolites such as phenolic compounds, and this mechanism could be the basis of the higher antioxidant capacities of plant extracts treated by nitrogen and ozone compounds [30].

Ozonation is a novel eco-friendly method that utilizes O_3_, a triatomic form of oxygen. O_3_ is a naturally occurring, highly reactive gas molecule that can be generated via electric discharge and/or UV radiation, either separately or in combination with NOx [31]. In the present study, ozonation has been employed to test the hypothesis of an enhancement of the antibacterial activity in the Mediterranean plant *Q. ilex*. As shown in Figure 4, O_3_ treatment caused an increase in antibacterial efficacy by lowered MIC values for some microbial strains such as *S. aureus* and *K. pneumoniae*. This suggests a possible effect of O_3_ to contrast some bacterial strains involved in common human diseases and identified as multi-drug resistant microorganisms. The influence of O_3_ on polyphenolic compounds has been supported by two different hypotheses. In the first case, the presence of O_3_ molecules triggers the production of phenolic compounds by inducing a partial breakdown of cell structure. These changes have the potential to improve the extraction process efficiency and aid in the release of specific phenolic compounds that are bound to the cell wall [32]. Moreover, the varying structural compositions of different phenolic compounds lead to differences in their tendency to accumulate within the cell wall [33]. 

On the other hand, the changes in phenolic compounds could be related to ozone-induced changes in enzymatic activities. Generally, the activation of pre-existing enzymes has been found to induce rapid accumulation of phenols [34]. The final concentration of phenols in ozone-treated foods is determined by the balance between these two processes. In our work, we tested the possible antimicrobial activity of *Q. ilex* extracts treated with ozone and nitrogen. The control of nitrogen metabolism is crucial for improving plants’ ability to resist stress, and influences almost all biological processes within them [35]. Glutamine synthase (GS) and glutamate synthase (GOGAT) are fundamental enzymes that play a central role in converting inorganic nitrogen into amides and amino acids inside cells, thus linking together carbon and nitrogen metabolisms [36]. Moreover, ammonium serves as an essential intermediate in the nitrogen metabolism of plants. It can be acquired from external sources via root uptake and a reduction in nitrate, or internally through the degradation of amino acids and the process of photorespiration [37]. Differences in amino acid content in response to N treatment were related to the absorption and assimilation of N and amino acid turnover; for example, proline synthesis [38]. Our results could be related to this mechanism and the treatment by nitrogen or a combination of nitrogen and ozone could influence the production of secondary bioactive compounds in plants available to signal molecules. Since the accumulation of these compounds in plants may promote an increased level of antioxidant properties and antimicrobial performance, further investigation is required. In conclusion, the application of ozone and nitrogen on *Q. ilex* could be the basis for stimulating the plant to produce secondary metabolites that can be used as antioxidant and antibacterial molecules.

## 4. Materials and Methods

### 4.1. Growth Conditions

Two-year-old seedlings of *Q. ilex*, obtained from the nursery of Aurunci Regional Park (Central Italy), were transplanted into 7L pots filled with a mixture of sand, turf, and perlite and placed in a `walk-in’ chamber facility at the Department of Environmental Biology, Sapienza University of Rome. Air temperature was maintained at 27.9 ± 1.8 °C during the day and at 22.7 ± 0.9 °C at night. The relative humidity was 61 ± 6.1%. In each chamber, a photosynthetic active radiation of approximately 700 μmol m^−2^ s^−1^ was provided for 12 h per day by using 6 metal halide lamps (1000 W; Philips HPI-T). In Fusaro et al., 2017 [10] further details about the experimental setup were reported.

### 4.2. Experimental Design

After an acclimatation period of about 30 days to the chamber environmental conditions, 20 plants were randomly divided into 4 experimental sets. Ten plants per species were assigned to the control chamber and thus randomly divided as follows: five plants were assigned to the control experimental set (CTRL), and five plants to the nitrogen addition experimental set (N). Ten plants per species were assigned to the fumigated chamber and thus randomly divided as follows: five plants to the O_3_ treatment (O_3_), and five plants to the interaction experimental set, treated with both N and O_3_ (O_3_+N). The N addition treatment was divided into 7 aliquots and applied throughout the experimental period as an aqueous solution. For this, 100 mL of deionized water was weekly added to each pot with different doses of ammonium nitrate (NH_4_NO_3_): 0 mg for C plants and 0.031 mg for N treatment. The final nitrogen dose was equal to 20 kg N ha^−1^ yr^−1^ based on the soil surface area. The ozone fumigation was started after five nitrogen additions when the cumulative dose was roughly equivalent to 14 kg N ha^−1^ yr^−1^, which falls in the upper limit of the threshold load currently indicated as critical for Mediterranean vegetation. The acclimation to the N addition phase and fumigation period lasted 30 days and 10 days, respectively. During the fumigation period, experimental sets were kept in the control chamber under filtered air (O_3_ = 0 to maximal 5.8 ppb). The O_3_ and O_3_N sets were placed in the fumigation chamber and exposed for 10 consecutive days to a mean hourly O_3_ concentration of 87.00 ± 0.5 ppb for 5 h per day simulating the concentration found in the Mediterranean rural area during the summer period [9]. The cumulative exposure was 2585.47 ppb h^−1^, expressed as AOT40, calculated by summing up all the exceedances of the hourly O_3_ concentration above 40 ppb during daylight hours [17]. O_3_ was generated in the fumigation chamber by flowing pure oxygen over a UV light source (Helios Italquartz, Milan, Italy), and then added to the chamber air inlet via a Teflon tube. The O_3_ concentration at plant height was continuously monitored using a photometric O_3_ detector (Model 205, 2B Technologies, Boulder, CO, USA).

### 4.3. Plants Extraction

For the preparation of *Q. ilex* extracts, 20 g of lyophilized leaves (16 leaves for each experimental set) were homogenized for 3 min with 1.5 volumes (*w*/*v*) of 60% food-grade acetone. Then, homogenates were centrifuged at 10,000× *g* for 15 min at 4 °C. The supernatants were collected and filtered through no. 42 Whatman paper, whereas the residue was re-extracted once more, following the procedure previously described. Two different supernatants were combined and subsequently dried using a rotary evaporator [39].

### 4.4. Determination of Total Phenolic Compounds, Hydroxycinnamic Acids, and Flavonoid Content

Total phenolic content was determined using the Folin–Ciocalteu assay [40,41]. Absorbances were measured at 765 nm. The results were expressed as milligrams of gallic acid equivalents per gram dry weight (dw) of *Q. ilex* leaf. Conversely, nitrite-molybdenum reagent was prepared by dissolving 10% (*m*/*v*) sodium molybdate and 10% (*m*/*v*) sodium nitrite in distilled water to determine the content of hydroxycinnamic acids (HCAs). To prepare a dilute sodium hydroxide solution, 8.3 g of sodium hydroxide was made up to 100 mL with purified water in a volumetric flask. The amount of HCAs was defined by UV absorption spectrophotometry in amounts equivalent to catechin at a wavelength of 505 nm after reaction with the nitrite-molybdenum reagent. Catechin was used as the standard of comparison, since after reaction with the nitrite-molybdenum reagent, extracts of the studied raw material types form an absorption maximum at 505 nm, as does rosmarinic acid [42]. Flavonoid contents have been measured as previously reported in Zhischen et al. 1999 with some modifications [41]. Briefly, plant materials were extracted with 20.0 mL of water/ethanol solution (60%) at room temperature for 60 min. Then, hydroalcoholic extract was transferred to a 10 mL volumetric flask and a volume of 2% AlCl_3_ was added. After 25 min, the absorbance was measured at 430 nm.

#### Determination of Ascorbic Acid and Reduced Glutation (GSH) Content

Ascorbic acid content in *Q. ilex* leaves was obtained by treating 20 mg of leaves with 100 µL of 6% trichloroacetic acid (TCA) solution. Then, all samples were centrifuged at 1000× *g* for 20 min. Supernatant was measured at wavelength 530 nm using U-Vis spectrophotometer by using the dinitrophenyl hydrazine reagent method [43]. GSH was measured using *Q. ilex* leaves treated by 6 % metaphosphoric acid (pH 2.8) for two hours. Then, plant extracts were centrifuged at 12,000× *g* for 15 min. The supernatant was incubated for an hour with 2-vinylpyridine as a reduced glutathione (GSH) blocker. Afterward, 100 mM PBS (pH 7.5), 5 mM EDTA, 0.2 mM NADPH, 0.6 mM DTNB, and 3 U GR from yeast were added to the reaction mixture. The quantity of glutathione was determined using a standard curve. The content of GSH was calculated by subtracting the content of GSSG from the total glutathione [44].

### 4.5. Antioxidant Activity

Since no single assay can accurately reflect the wide range of antioxidant properties in a mixed or complex system, antioxidant activity is better appraised by at least two complementary methods. Two antioxidant assays, based on either 2,2-diphenyl-1-picrylhydrazyl assay (DPPH) or 2,20-azino-bis (3-ethylbenzothiazoline-6-sulphonic acid) (ABTS), were applied to evaluate the antioxidant properties of *Q. ilex*. ABTS and DPPH methods have been performed as previously reported in Verrillo et al., 2023 [41]. Briefly, for the preparation of samples, 100 µL of plant extracts were mixed by 1.9 mL of the ABTS working solution. The mixture was shaken for 2 min in the dark and its absorbance measured at 734 nm by a Perkin Elmer Lambda 25 UV/Vis Spectrometer (Waltham, MA, USA). Results were expressed as Trolox Equivalent Antioxidant Capacity (TEAC) based on a linear calibration curve of Trolox. Conversely, DPPH (2,2-diphenyl-1-picrylhydrazyl) free-radical scavenging assay to determine antioxidant capacity has been carried out mixed 50 μL of plants extract to 150 μL of a 0.3 mM DPPH solution in ethanol. In comparison, a blank was prepared by adding 50 μL of Milli Q water to the DPPH ethanol solution. The mixture was vortexed for 2 min, incubated in the dark for 60 min at room temperature and its absorbance was measured at 517 nm by a Perkin Elmer Lambda 25 UV/Vis Spectrometer. Small absorbance values indicated large free radical scavenging activity (RSA), which was calculated as a percentage of DPPH inhibition using the following equation: % RSA = {[(A_0_ − A_i_)/A0] × 100}, where A0 is the absorbance of the control solution and Ai that of the plant extracts. All measurements were performed with three dilutions. Ferric reducing antioxidant potential assay was and carried out by mixing an aliquot (200 μL) of an extract to 3 mL of FRAP reagent (10 parts of 300 mM sodium acetate buffer at pH 3.6, 1 part of 10 mM TPTZ solution and 1 part of 20 mM FeCl_3_·6H_2_O solution). The reaction mixture was incubated in a water bath at 37 °C. The increase in absorbance at 593 nm was measured at 30 min. The antioxidant capacity based on the ability to reduce ferric ions of the extract was expressed as μmol Trolox equivalents per gram of plant material on a dry basis.

### 4.6. Antimicrobial Activity

The antibacterial activity of plant extracts was carried out by broth microdilution method (MIC) in a Mueller Hinton Broth medium by using sterile 96-well polypropylene microtiter [41]. Antibacterial screening was performed by testing two gram-positive (*Staphylococcus aureus* ATCC6538P; *Enterococcus faecalis* ATCC29212) and three gram-negative bacterial strains (*Escherichia coli* ATCC35218; *Pseudomonas aeruginosa* ATCC27355; *Klebsiella pneumoniae* ATCC700503). Briefly, twofold serial dilutions of different plant extracts were carried out in the test wells to obtain concentrations ranging from 1 to 1000 μg·mL^−1^ starting from an initial concentration of 2000 mg·mL^−1^. Then, the bacterial cells were inoculated from an overnight culture at a final concentration of about 5 × 10^5^ CFU mL^−1^ and incubated with a different sample overnight at 37 °C. Finally, the MIC values were estimated by measuring the spectrophotometric absorbance of microtiter plates at 570 nm. Moreover, the smallest concentration at which no turbidity was observed was considered the MIC value. Furthermore, albumin serum bovine (BSA), a protein without antibacterial properties, and ampicillin, a common antibiotic, were used as a negative and positive control, respectively. All oak samples were tested in triplicate and performed by three independent experiments.

### 4.7. Statistical Analysis

The significant difference between mean values was determined by the one-way analysis of variance (ANOVA), and the means (n = 3) were validated by applying Tukey’s test at the 0.05 significance level by using the XLSTAT software (Addinsoft, v. 2014).

## 5. Conclusions

Our novel work investigated how exposure to O_3_ and N depositions, as well as the flavonoid, hydrocycinammic, and phenolic contents, affect antioxidant and antibacterial activities in *Q. ilex* leaves extracts, through a controlled experiment. The results showed that the interaction between treatments upregulates the antioxidant and antimicrobial activity of the extracts against some pathogenic bacterial strains. This process could be influenced by the production of several bioactive compounds in plants, which are available for signaling molecules. Furthermore, the multiple exposure to O_3_ and N has proven to be an effective condition to induce a massive stimulation of phenolic moieties related to shikimic acid biochemical pathways in *Q. ilex*. The functional groups of these molecules can be used as antioxidant and antibacterial molecules, suggesting the potential employment of these natural extracts in nutraceutical fields.

## Figures and Tables

**Figure 1 plants-13-01154-f001:**
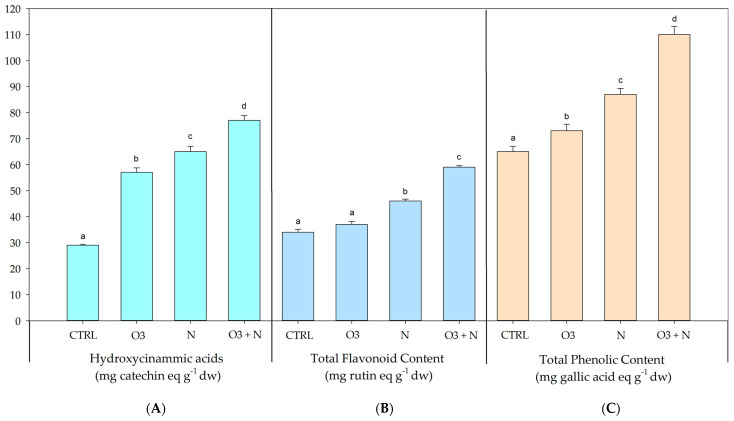
Spectrophotometric determinations of (panel **B**) total flavonoid content (mg rutin eq. g^−1^ dw), (panel **A**) hydroxycinammic acids (mg catechin equivalent g^−1^ dw), and (panel **C**) total phenolic content (mg gallic acid eq. g^−1^ dw) in the different experimental sets: *Q. ilex* control leaves samples (Qi-CTRL), treated only with O_3_ (Qi-O_3_), only with N (Qi-N), and with O_3_ plus N (Qi O_3_+N). Data are presented as means ± s.d. from three independent replicates. Different letters indicate significant differences between groups, *p* ≤ 0.05 for ANOVA and Tukey’s post hoc test (*p* ≤ 0.05).

**Figure 2 plants-13-01154-f002:**
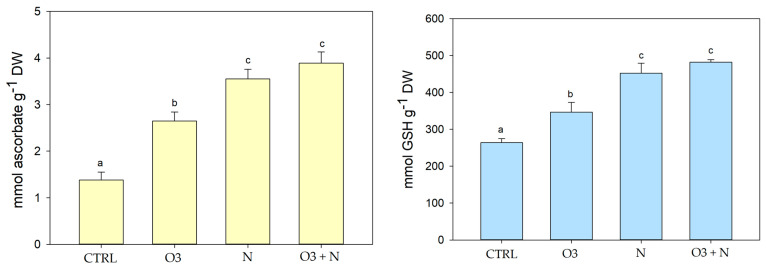
Contents of reduced glutathione (mmol GSH g^−1^ DW) and ascorbic acid (mmol ascorbate g^−1^ DW) in the different experimental sets: *Q. ilex* control leaves samples (Qi-CTRL), treated only with O_3_ (Qi-O_3_), only with N (Qi-N), and with O_3_ plus N (Qi O_3_+N). Data are presented as means ± s.d. from three independent replicates. Different letters indicate statistical significance.

**Figure 3 plants-13-01154-f003:**
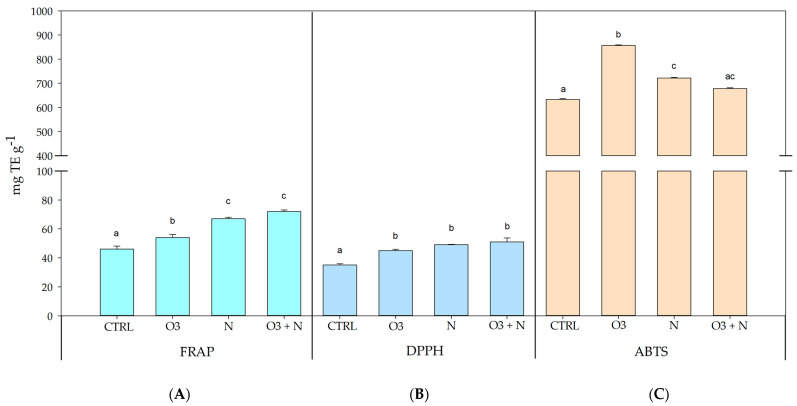
Antioxidant activity evaluated by (panel **B**) DPPH assay, (panel **A**) FRAP method, and (panel **C**) ABTS of the different experimental sets: *Q. ilex* control leaves (Qi-CTRL), treated only with O_3_ (Qi-O_3_), only with N (Qi-N), and with O_3_ plus N (Qi O_3_+N). All the assays are expressed as mg Trolox equivalent g^−1^. Data are presented as means ± s.d. from three independent replicates. Different letters indicate significant differences between groups, *p* ≤ 0.05 for ANOVA and Tukey’s post hoc test (*p* ≤ 0.05).

**Figure 4 plants-13-01154-f004:**
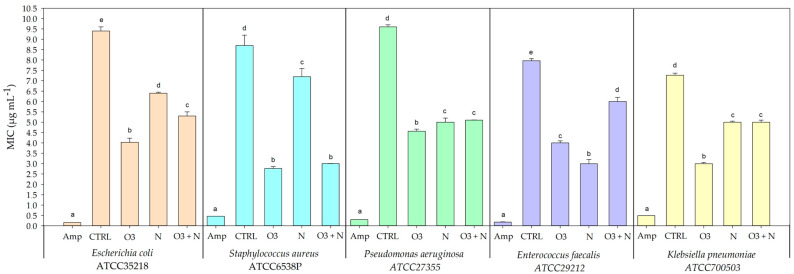
Antibacterial activity as MIC (Minimal Inhibitory Concentration) expressed as µg mL^−1^ against some MDR gram-positive and gram-negative bacterial strains. Assays were carried out by the broth dilution method in Muller Hinton Broth. Data are presented as means ± s.d from three independent replicates. Different letters indicate significant differences between groups for ANOVA and Tukey’s post hoc test (*p* ≤ 0.05).

## Data Availability

The data are contained within the article and are available upon request.

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
