# Peer review of "Oxidative Stress Response Mechanisms Sustain the Antibacterial and Antioxidant Activity of Quercus ilex"

_plants, 2024, doi:10.3390/plants13081154_

Round 1
Reviewer 1 Report (Previous Reviewer 1)
Comments and Suggestions for Authors
Thank you for the response. Still, the manuscript needs extensive changes. The total phenolic compounds, hydroxycinammic acids, and flavonoid contents are not good enough to show antioxidant activities. Authors are suggested to add at least data on ascorbic acid and glutathione, which are known for potential activity against abiotic stresses.
Author Response
Reviewer 1
The manuscript entitled “Effects of ozone and nitrogen supply on the chemical composition, antibacterial and antioxidant activity in leaves extract of Q.ilex L” is well written. However, few major questions need to be addressed. I recommend resubmitting this manuscript after comprehensive changes in the manuscript.
R1: Thank you for the response. Still, the manuscript needs extensive changes. The total phenolic compounds, hydroxycinammic acids, and flavonoid contents are not good enough to show antioxidant activities. Authors are suggested to add at least data on ascorbic acid and glutathione, which are known for potential activity against abiotic stresses.
A: We recognize the value of the Reviewer’s suggestion, and we added the evaluation of ascorbic acid and glutation in this manuscript (Lines 129-146). These data confirmed the trend observed in all other experimental analysis.
Reviewer 2 Report (Previous Reviewer 2)
Comments and Suggestions for Authors
The manuscript was revised based on the comments provided and is acceptable.
Minor comment:
I don't think capital letters are needed in names of chemical compounds, like Total Flavonoid Content (190., 191 L) ; Different letters (187 L) etc. Please correct.
Please use Q. ilex extracts or leaves against Quercus extracts.
Author Response
Reviewer 2
The manuscript was revised based on the comments provided and is acceptable.
Minor comment:
I don't think capital letters are needed in names of chemical compounds, like Total Flavonoid Content (190., 191 L) ; Different letters (187 L) etc. Please correct.
Please use Q. ilex extracts or leaves against Quercus extracts.
A: Accordingly with Reviewer’s comments we modified the capital letters of all chemical compounds and use in all manuscripot Q.ilex extracts.
Reviewer 3 Report (Previous Reviewer 3)
Comments and Suggestions for Authors
In general the authors have improved the work since the previous version. However, there are some shortcomings, which I list below:
- There are results that are not reflected in the figure. Figure 1 does not show that the combination of treatments is the best option as stated in line 95.
- In figure 1 (third panel) the bar "ac" appears in the bar "ac" this does not make sense if in the rest of the bars the c is not in solitary. This statistical analysis needs to be revised. Also with these error bars O3 and N should be different. On the other hand, in the FRAP, CTRL and O3 do not seem to indicate that the differences are statistically significant.
- Abbreviations in figure legends must be defined.
- The conclusion of line 316 is not supported by the results shown.
- The merging of results and discussion dilutes the results obtained in the study.
-More experiments are needed to be able to say the things that are addressed in the discussion. And progress needs to be made so that the work is not merely descriptive. It would be crucial to identify some compounds by more precise techniques (HPLC).
Also some formatting errors remain, such as double spacing (line 315), colons (line 316), the title 4.5 has some brackets that should not be there or some missing information, the centrifugation speed should be indicated in g and not in rpm.
Author Response
In general the authors have improved the work since the previous version. However, there are some shortcomings, which I list below:
R3: There are results that are not reflected in the figure. Figure 1 does not show that the combination of treatments is the best option as stated in line 95.
A: Thank you for the observation. We modified the Figure and adding an other figure we change their numeration.
R3: In figure 1 (third panel) the bar "ac" appears in the bar "ac" this does not make sense if in the rest of the bars the c is not in solitary. This statistical analysis needs to be revised. Also with these error bars O3 and N should be different. On the other hand, in the FRAP, CTRL and O3 do not seem to indicate that the differences are statistically significant.
A: We apologise for the misprint. We carefully revised the manuscript. Thank you for the suggestions
R3: Abbreviations in figure legends must be defined.
A: Thank you for the observation. We defined the abbreviation in all Figures.
R3: The conclusion of line 316 is not supported by the results shown.
A: Accordingly with Reviewer’ comment we modified the conclusion.
R3: The merging of results and discussion dilutes the results obtained in the study.
A: Following the Reviewer’s suggestion, we splitted results and discussion in the manuscript
R3: More experiments are needed to be able to say the things that are addressed in the discussion. And progress needs to be made so that the work is not merely descriptive. It would be crucial to identify some compounds by more precise techniques (HPLC).
A: We understand the Reviewer’s point of view but we decided to carry on this approach since, we focused our metabolic profiling on functional groups because the manuscript aims to identify, for the first time, the potential antimicrobial and antioxidant activities of Quercus samples subjected to different types of abiotic stresses. For this reason, we did not perform analyses to understand the specific composition of vegetable extracts focusing on the presence and the role of principal biochemical functional classes of bioactive compounds and their relation to antioxidant and antimicrobial activities of vegetable extracts. Additionally, we added the evaluation of ascorbic acid and glutathione content to improve the bioactive compounds investigated in this manuscript suggested to R1
R3: Also some formatting errors remain, such as double spacing (line 315), colons (line 316), the title 4.5 has some brackets that should not be there or some missing information, the centrifugation speed should be indicated in g and not in rpm. Some general comments for improvement:
A: We apologise for the misprint. We carefully revised the manuscript. Thank you for the suggestions.
Round 2
Reviewer 1 Report (Previous Reviewer 1)
Comments and Suggestions for Authors
The authors have incorporated all the suggestions properly. I recommend this manuscript to publish in the current version.
Author Response
R1: The authors have incorporated all the suggestions properly. I recommend this manuscript to publish in the current version.
A: We are grateful to the reviewer for the thorough revision process.
Reviewer 3 Report (Previous Reviewer 3)
Comments and Suggestions for Authors
Having reviewed the article, I see that the authors have made some of the requested changes, but there are still some that have not, especially some concerning the statistics in figure 3.
On the other hand, some minor changes are necessary, as there are still a large number of typographical errors.
Check that abbreviations, e.g. DW (dry weight) appears the first time it is named and then only the abbreviation. The same is true for Quercus ilex, put Q. ilex throughout the manuscript.
Unify nomenclature, mg/g DW or mg g-1 DW.
Figure 3- in the FRAP the error bar of the control and O3 overlap, how has the statistical test been done? The same is true for the DPPH. This was already said in the previous review.
It would be interesting to add the hypothesis of the study in the last paragraph of the introduction.
Figure 4- decimals should be separated by "." and not by ",".
Line 189- there is no space after the comma.
Line 429- correct "eachexperimental set"
Line 209 - space between brackets((μg/mL)(Figure))
Line 217 - add a full stop at the end of the sentence.
Line 220 - change , by . (μg/mL),
Line 277- remove full stop at the beginning of the sentence
Line 286 - remove double dot
Line 292 - remove space before full stop
Line 294 - remove space before comma
Line 299 - remove double dot
Line 301 - remove double dot
Line 309-318 - this information is not relevant when writing a discussion.
Line 314-318 - different letter size.
Line 328 - add space after [25]
Line 347 - O3 with superscript?
Line 359 - remove full stop after "wall."
Line 361 - remove full stop after "wall."
Line 385 - What is this?
Line 458 - space in Q.ilex after full stop
Line 510 - would be paragraph 4.8
Line 522 - O3 son subindice, correir.
Line 518 - For some strains of bacteria the combination does not have a major effect, so the conclusion of this line is not entirely correct.
Author Response
R3: Having reviewed the article, I see that the authors have made some of the requested changes, but there are still some that have not, especially some concerning the statistics in figure 3. On the other hand, some minor changes are necessary, as there are still a large number of typographical errors.
A: We are grateful to the reviewer for the thorough revision process. We have implemented the reviewer’s observations
R3: Check that abbreviations, e.g. DW (dry weight) appears the first time it is named and then only the abbreviation. The same is true for Quercus ilex, put Q. ilex throughout the manuscript.
A: We modified all manuscript adding correct abbreviations.
R3: Unify nomenclature, mg/g DW or mg g-1 DW.
A: The nomenclature have been revised.
R3: Figure 3- in the FRAP the error bar of the control and O3 overlap, how has the statistical test been done? The same is true for the DPPH. This was already said in the previous review.
A: Thank you for spotting the error. Mistakenly the error bars were set to 5 in the graph designer software, the figure is now graphically correct.
R3: It would be interesting to add the hypothesis of the study in the last paragraph of the introduction.
A: The hypothesis has been more clearly outlined in the last intro paragraph
Line 80-82 In this work, we hypothesize that the employment of O3 and N excess could presumably elicit better antibacterial and antioxidant activities in the leaf extracts of the evergreen species Q. ilex.
R3: Figure 4- decimals should be separated by "." and not by ",". The decimal separators have been corrected
Line 189- there is no space after the comma.
Line 429- correct "eachexperimental set"
Line 209 - space between brackets((μg/mL)(Figure))
Line 217 - add a full stop at the end of the sentence.
Line 220 - change , by . (μg/mL),
Line 277- remove full stop at the beginning of the sentence
Line 286 - remove double dot
Line 292 - remove space before full stop
Line 294 - remove space before comma
Line 299 - remove double dot
Line 301 - remove double dot
Line 309-318 - this information is not relevant when writing a discussion. Trovo difficile individuarlo
Line 314-318 - different letter size.
Line 328 - add space after [25]
Line 347 - O3 with superscript?
Line 359 - remove full stop after "wall."
Line 361 - remove full stop after "wall."
Line 385 - What is this? Spero intenda che si deve riparafrasare tra 384 e 385
Line 458 - space in Q.ilex after full stop
Line 510 - would be paragraph 4.8
Line 522 - O3 son subindice, correir.
Line 518 - For some strains of bacteria the combination does not have a major effect, so the conclusion of this line is not entirely correct.
A: All the following issues have been thoroughly solved, and the manuscript has been revised
This manuscript is a resubmission of an earlier submission. The following is a list of the peer review reports and author responses from that submission.
Round 1
Reviewer 1 Report
Comments and Suggestions for Authors
Reviewer’s Comments
The manuscript entitled “Effects of ozone and nitrogen supply on the chemical composition, antibacterial and antioxidant activity in leaves extract of Q.ilex L” is well written. However, few major questions need to be addressed. I recommend resubmitting this manuscript after comprehensive changes in the manuscript.
Why authors have not perform the metabolic profiling of the compounds? Only total phenolic compounds, hydroxycinammic acids and flavonoids contents are not enough to properly understand the composition of the compounds in the extract.
To examine the time-dependent effect on the experiment is not designed, accordingly?
On which basis, bacteria were selected, incorporate in the manuscript.
Compound activity is not enough for the bacteria, authors should add pictures related to the activity of the composition on the bacteria.

Reviewer 2 Report
Comments and Suggestions for Authors
The manuscript “Effects of ozone and nitrogen supply on the chemical composition, antibacterial and antioxidant activity in leaves extract of Q.ilex L“ is dedicated to a specific and relevant topic and is potentially publishable. Unfortunately the research presented is insufficiently substantiated, contains confusion and uncertainty that needs to be clarified and supplemented.
Please follow the comments:
What species was studied? Q. ilex L. does not provide any information. The name of the genus must be written in full in the manuscript title, abstract and text (the first time it is mentioned).
The Introduction should reveal the problem of the presented research. In general the problem was considered, and it is not clear why such a study is necessary. Way was selected the corresponding species (Q. ilex (?)? What was its novelty?
The submitted manuscript mainly presented considerations on the previous studies by other authors. The research results seem weak.
The study design suggests that two separate treatments were used and then combined together. The results of their interaction should be revealed using statistical methods.
The title of figures 1 and 2 should be corrected. Incorrect expression ‘without treatments’.
Figs 1 and 2 have been redundantly divided into separate parts (A, B, C) which can be combined into one or the results can be tabulated.
Please use the accepted dimensional abbreviation (milligram of gallic acid equivalents per gram dry weight (dw)).
The description of experimental design is highly confused. How many species have you treated, as you write “20 plants per species” etc.
How many plants you have treated?
What plant material and how was collected for further analyses?
How many repetitions of the material were collected?
The study design suggests that in addition a Two Way Anova should be used.
Why for ABTS and DPPH methods were abbreviations used, and for FRAP the full name?
All Latin names must be written in italic.
Pleas reconsider understandably ,,two different steps. Finally, the two supernatants have been combined to obtain the sample for biological characterization” (24-25).
Please make it clear: Flavonoid content has been carried out hax been previously reported with some modifications (258)
Conclusions must be reconsidered.
What did the Authors do new?
Reviewer 3 Report
Comments and Suggestions for Authors
The present study (“Effects of ozone and nitrogen supply on the chemical composition, antibacterial and antioxidant activity in leaves extract of Q.ilex L”) submitted by Mariavittoria Verrillo et al., could be an interesting work, as they seek to understand the effect that N and ozone have on extracts that could have interesting biotechnological applications, but in my opinion the work falls short of being published in a high impact journal, as it only makes a general characterisation, it would be necessary to add some more experiments and go deeper into the characterisation of the extracts, identifying secondary metabolites of interest that could provide an answer to the changes found. I therefore encourage the authors to take the work back and try to deepen it and then resubmit it.
Some general comments for improvement:
· - The title simply describes what has been analysed, it should indicate something more relevant to the results obtained that will provoke interest in the reader.
· - Abstract is poorly written. The abstract begins by describing the materials and methods of the study without introducing the topic and stating an objective.
· - In the key words, ozone is mentioned and not nitrogen, when according to the title both would be at the same level of "importance".
· There are many spelling mistakes, superscripts not in place (e.g. line 298; “-1”), scientific names do not appear in italics, words joined without a space (e.g. line 48), line 27 [xx],…
· The concluding paragraph is redundant, as the last two sentences are repetitive, as well as highly speculative, because they suggest that it is due to the production of secondary metabolites, but none have been measured directly by specific and quantitative techniques such as HPLC-MS.
· In addition to deepening the characterisation of the extract, information should be provided on the physiological state of the plants, biomass measurements, chlorophylls, gas exchange measurements, etc., which are key to understanding how the plants are reacting to the treatments and even to determine whether it represents a level of stress.
· In the discussion the whole route of synthesis of phenolic compounds is discussed, perhaps a metabolomic determination of these elements in the plants after the treatments should be carried out.
Comments on the Quality of English Language
I don't feel fully qualified to proofread the English but it does need proofreading and there are quite a few typos.